# *HSPA8* Single-Nucleotide Polymorphism Is Associated with Serum HSC70 Concentration and Carotid Artery Atherosclerosis in Nonalcoholic Fatty Liver Disease

**DOI:** 10.3390/genes13071265

**Published:** 2022-07-16

**Authors:** Wenli Zhao, Hitoe Mori, Yuki Tomiga, Kenichi Tanaka, Rasheda Perveen, Keiichiro Mine, Chika Inadomi, Wataru Yoshioka, Yoshihito Kubotsu, Hiroshi Isoda, Takuya Kuwashiro, Satoshi Oeda, Takumi Akiyama, Ye Zhao, Iwata Ozaki, Seiho Nagafuchi, Atsushi Kawaguchi, Shinichi Aishima, Keizo Anzai, Hirokazu Takahashi

**Affiliations:** 1Division of Metabolism and Endocrinology, Faculty of Medicine, Saga University, Saga 849-8501, Japan; 21624009@edu.cc.saga-u.ac.jp (W.Z.); 16624022@edu.cc.saga-u.ac.jp (H.M.); sw0864@cc.saga-u.ac.jp (Y.T.); sj8833@cc.saga-u.ac.jp (K.T.); rashu_bcmb@yahoo.com (R.P.); sv7899@cc.saga-u.ac.jp (K.M.); chlkqiko@gmail.com (C.I.); sailingxyz94@yahoo.co.jp (W.Y.); y.05211027@gmail.com (Y.K.); f8451@cc.saga-u.ac.jp (T.K.); akiyamat@cc.saga-u.ac.jp (T.A.); ozaki@cc.saga-u.ac.jp (I.O.); su2733@cc.saga-u.ac.jp (S.N.); akeizo@cc.saga-u.ac.jp (K.A.); 2Liver Center, Saga University Hospital Faculty of Medicine, Saga University, Saga 849-8501, Japan; e6140@cc.saga-u.ac.jp (H.I.); ooedasa@cc.saga-u.ac.jp (S.O.); 3Division of Mucosal Immunology, Research Center for Systems Immunology, Medical Institute of Bioregulation, Kyushu University, Fukuoka 812-8582, Japan; 4Innovative Institute of Chinese Medicine and Pharmacy, Shandong University of Traditional Chinese Medicine, Jinan 250014, China; zhaoye@sdutcm.edu.cn; 5Health Administration Centre, Saga Medical School, Saga University, Saga 849-8501, Japan; 6Education and Research Center for Community Medicine, Faculty of Medicine, Saga University, Saga 849-8501, Japan; akawa@cc.saga-u.ac.jp; 7Department of Pathology and Microbiology, Faculty of Medicine, Saga University, Saga 849-8501, Japan; saish@cc.saga-u.ac.jp

**Keywords:** cardiovascular disease, liver biopsy, carotid artery ultrasound, precision medicine

## Abstract

There is an association between nonalcoholic fatty liver disease (NAFLD) and atherosclerosis, but the genetic risk of atherosclerosis in NAFLD remains unclear. Here, a single-nucleotide polymorphism (SNP) of the heat shock 70 kDa protein 8 (*HSPA8*) gene was analyzed in 123 NAFLD patients who had been diagnosed using a liver biopsy, and the NAFLD phenotype including the maximum intima–media thickness (Max-IMT) of the carotid artery was investigated. Patients with the minor allele (A/G or G/G) of rs2236659 showed a lower serum heat shock cognate 71 kDa protein concentration than those with the major A/A allele. Compared with the patients with the major allele, those with the minor allele showed a higher prevalence of hypertension and higher Max-IMT in men. No significant associations between the *HSPA8* genotype and hepatic pathological findings were identified. In decision-tree analysis, age, sex, liver fibrosis, and *HSPA8* genotype were individually associated with severe carotid artery atherosclerosis (Max-IMT ≥ 1.5 mm). Noncirrhotic men aged ≥ 65 years were most significantly affected by the minor allele of *HSPA8*. To predict the risk of atherosclerosis and cardiovascular disease, *HSPA8* SNP genotyping might be useful, particularly for older male NAFLD patients.

## 1. Introduction

Nonalcoholic fatty liver disease (NAFLD) is the most common cause of chronic liver disease, affecting approximately 30% of adults worldwide [1]. By 2030, NAFLD is predicted to become the most common etiological factor behind the need for liver transplantation in Western countries [2,3]. In addition to cirrhosis-related complications, liver cancer, and extrahepatic cancer, recent epidemiological studies confirmed that cardiovascular disease (CVD) is a main cause of death in NAFLD patients, and hepatic fibrosis caused by NAFLD is associated with both liver-related mortality and CVD events [4,5,6,7]. NAFLD is also associated with several well-known metabolic risk factors for CVD and atherosclerotic CVD (ASCVD), including obesity, insulin resistance, metabolic syndrome, type 2 diabetes, and dyslipidemia [8]. To date, no clinical management strategy to predict a particular risk for future ASCVD to distinguish between liver-related events and liver cancer in NAFLD has been established [9].

Carotid intima–media thickness (IMT) is a surrogate marker for the presence and progression of atherosclerosis [10,11]. IMT is also a strong predictor of future cerebral and cardiovascular events [12,13]. Moreover, recent studies showed that atherosclerosis is also associated with NAFLD. A meta-analysis by Sookoian et al. revealed that atherosclerosis that was evaluated using the carotid IMT was more severe in NAFLD patients than in those without NAFLD [14]. The presence of carotid plaques was also more common in patients with NAFLD than in those without it [14,15]. Moreover, NAFLD was associated with the progression of coronary artery atherosclerosis independently of known risk factors, including age, hypertension, and diabetes, and NAFLD with a risk of advanced fibrosis was found to strongly promote atherosclerosis formation in the coronary artery [16]. These findings suggest that NAFLD per se and hepatic fibrosis are associated with atherosclerosis and an increased risk of ASCVD.

The genetic risk factors that can predict ASCVD in NAFLD are not completely understood. Recent genome-wide association studies (GWAS) showed that the single-nucleotide polymorphism (SNP) of patatin-like phospholipase domain-containing protein 3 (*PNPLA3*) rs738409 was the most significant genetic risk factor for steatosis, fibrosis, and hepatocellular carcinoma in NAFLD [17,18]. The association between this SNP and CVD has also been studied. Petta et al. reported that carotid IMT was greater in those with *PNPLA3* G/G than in those with C/C or C/G [19]. However, Di Costanzo et al. reported that participants with *PNPLA3* C/C showed similar carotid IMT to those with the *PNPLA3* G/G genotype in blood donor volunteers [20]. In addition, Unalp-Arida and Ruhl examined the relationship between NAFLD and mortality in 5662 participants who were genotyped for *PNPLA3* I148M (NCBI dbSNP ID: rs738409 C>G). They identified that M variant heterozygosity was a significant risk factor for cardiovascular disease [21]. Thus, the association between the *PNPLA3* genotype and CVD in NAFLD has not been confirmed. 

Heat shock cognate 71 kDa protein (HSC70) is a member of the heat shock protein 70 (HSP70) family that is encoded by the heat shock protein family A member 8 (*HSPA8*) gene. It is a molecular chaperone involved in many cellular processes [22]. Recent experimental and clinical studies have shown the association between HSC70/*HSPA8* and CVD. The HSC70 chaperone system mediates cardiac myosin binding protein C turnover and is associated with hypertrophic cardiomyopathy [23]. *HSPA8* genetic variants are associated with the risk of coronary heart disease [24]. However, to the best of our knowledge, no study has yet investigated the association between *HSPA8* SNP and atherosclerosis in NAFLD. The current study thus investigated the association among an *HSPA8* SNP, atherosclerosis, and hepatic pathogenesis in NAFLD. 

## 2. Materials and Methods

### 2.1. Patients

We analyzed data from 123 Japanese patients with a histological diagnosis of NAFLD at Saga University Hospital. Patients with habitual alcohol intake (ethanol consumption > 210 g/week for men and >140 g/week for women) or a positive hepatitis B surface antigen and/or a positive hepatitis C antibody test result were not included. Patients with a diagnosis of autoimmune liver disease, drug-induced hepatotoxicity, hemochromatosis, Wilson’s disease, or endocrine disease were not included. The study protocol was approved by the Clinical Research Ethics Review Committee at Saga University Hospital (approval number: 2017-01-12) and the Institutional Review Board from the Faculty of Medicine, Saga University (2019-30-37). The study was conducted in accordance with the principles of the 1975 Declaration of Helsinki, as revised in 2013. Written informed consent was obtained from all participants in this study.

### 2.2. Physical Examination and Serum Biochemical Measurements

Clinical data including age, sex, and medical history of comorbid diseases were collected from the patients’ medical records. Body weight and height were measured on the day of liver biopsy, and body mass index (BMI) was calculated as body mass (kg) divided by the square of height (m^2^). Venous blood samples were obtained after an overnight fast and used to measure the platelet count, and the albumin, aspartate aminotransferase, alanine aminotransferase (ALT), γ-glutamyl transpeptidase, alkaline phosphatase, fasting plasma glucose (FPG), glycated hemoglobin A1c (HbA1c), insulin, total cholesterol, high-density-lipoprotein cholesterol (HDL-C), low-density-lipoprotein cholesterol (LDL-C), triglyceride, blood urea nitrogen (BUN), and creatinine (Cr) levels were determined using conventional laboratory techniques. Insulin resistance—using the homeostatic model assessment for insulin resistance (HOMA-IR)—was determined as fasting insulin (IU/mL) × FPG (mg/dL)/405. The estimated glomerular filtration rate (eGFR) was calculated using the following equation: 194 × Cr (mg/dL) − 1.094 × age (years) − 0.287 for men, and the values obtained for women were multiplied by 0.739 [25]. The HSC70 concentration in 123 serum samples was determined using an enzyme-linked immunosorbent assay (ELISA) kit (#SKT-106; StressMarq Biosciences Inc., Victoria, BC, Canada) in accordance with the manufacturer’s protocol. The minimum detection level of the ELISA kit was 1.54 ng/mL. All of the samples analyzed in the present study showed a concentration higher than this.

### 2.3. Liver Biopsy and Histological Assessment

An ultrasonography-guided liver biopsy was performed using a 16-gauge biopsy needle. Liver biopsy sections were stained with hematoxylin and eosin and Azan stain and evaluated by an experienced pathologist (S.A.) who specializes in liver pathology and was blinded to the clinical data. Hepatic steatosis, lobular inflammation, and hepatocyte ballooning were evaluated using the NAFLD activity score (NAS) [26]. Liver fibrosis was classified in accordance with the work of Kleiner et al. [26] and Brunt et al. [27]. Steatosis was scored as 0, 1, 2, or 3 (score 0, <5%; score 1, 5%–33%; score 2, 34%–66%; and score 3, >66% of the biopsy), and fibrosis was scored as 0, 1, 2, 3, or 4 (stage 0, no fibrosis; stage 1, perisinusoidal or periportal fibrosis; stage 2, perisinusoidal and portal/periportal fibrosis; stage 3, bridging fibrosis; and stage 4, cirrhosis). NAFLD was diagnosed if the steatosis score was 1 or higher. Patients who presented with at least grade 1 of each of the three features (steatosis, ballooning, and lobular inflammation) and had a compatible overall pattern of liver injury were classified as having nonalcoholic steatohepatitis (NASH), and the remaining patients were diagnosed with NAFL (NAFLD without NASH) [28].

### 2.4. Intima–Media Thickness Measurements

The carotid artery IMT was measured using ultrasound (LOGIQ E10; GE Healthcare Japan, Tokyo, Japan). IMT was measured on the right and left sides in the areas of the common carotid artery, bulbus, and internal carotid artery using a linear probe (#L2-9-D; GE Healthcare Japan, Tokyo, Japan). The maximum-IMT (Max-IMT) was defined as the maximum difference between the first (intima–lumen) and second (media–adventitia) interfaces on the far wall of either the right or the left carotid artery bulb, internal carotid artery, and common carotid artery [29,30]. Max-IMT was expressed in mm, and the carotid plaque was included in the analysis [14,29]. Max-IMT ≥ 1.5 mm was considered to confer a significant risk for future cardiovascular events [31].

### 2.5. DNA Extraction and HSPA8 Genotyping

To extract DNA from whole blood, an automatic nucleic acid extraction system (Quick Gene DNA whole blood kit S, Cat. No. DB-S; Fujifilm, Tokyo, Japan) was used, in accordance with the manufacturer’s protocol. The concentration of DNA was measured for all of the samples and then adjusted to 50 ng/µL with double-distilled water. The purity of DNA was checked using a spectrocolorimeter (NanoDrop 2000 C; Thermo Fisher Scientific, Waltham, MA, USA) and the absorbance ratio of 260 nm/280 nm was between 1.8 and 1.9 for all of the DNA samples. The *HSPA8* sequence was in accordance with the NCBI Reference Sequence (NCBI ClinVar ID: 3312) can be found at https://www.ncbi.nlm.nih.gov/gene/3312 (accessed on 25 May 2022), and NC_000011.10: g.123062473A>G (GRCh38) (rs2236659) was investigated in the present study. The genomic region containing the SNP was amplified using polymerase chain reaction (PCR) with a forward primer (5’-GCCAAGAAGCCGAATCTGTT-3’) and a reverse primer (5’-GCCTCACGATAACGCACTCA-3’). The PCR conditions were as follows: initial denaturation at 96 °C for 2 min; then 40 cycles of 96 °C for 20 s, 64 °C for 20 s, and 72 °C for 20 s; followed by a final extension step at 72 °C for 2 min. The PCR product size was verified using agarose gel electrophoresis. After the PCR product had been refined using the ExoSAP-IT (Thermo Fisher Scientific), the sequencing reaction was performed using a sequencing primer (5’-GCCCAAACCCCTCCCTTCAG-3’) in a Bio-Rad DNA Engine Dyad PTC-220 Peltier Thermal Cycler using ABI PRISM^®^ BigDye^®^ Terminator v3.1 Cycle Sequencing Kits (Applied Biosystems, Waltham, MA, USA), in accordance with the manufacturer’s protocol. The genotype was defined as major allele (homozygous for the major allele; A/A) or minor allele (heterozygous or homozygous for the minor allele; A/G or G/G). For the genotyping of *PNPLA3*, a predesigned TaqMan probe (Applied Biosystems, Foster City, CA, USA) was used for the genotyping of rs738409 (C_7241_10), in accordance with the manufacturer’s protocol.

### 2.6. Statistics

Continuous variables were expressed as the mean ± standard deviation (SD) or as the median (interquartile range). Qualitative data were presented as number and percentage, which was indicated in parentheses. A chi-squared test and Fisher’s exact test were used for categorical factors. Numerical variables were compared using the Mann–Whitney U test. Spearman’s correlation coefficient was used to test the correlation between variables. The Kruskal–Wallis test was used to test the differences between two or more groups. A decision-tree algorithm was performed to investigate the profiles of patients with significant atherosclerosis (Max-IMT ≥ 1.5 mm), including the *HSPA8* genotype, age, sex, and hepatic fibrosis stage. Statistical analyses were performed using SPSS version 22.0 software (IBM Corp., Armonk, NY, USA) and GraphPad 9 software (GraphPad, San Diego, CA, USA). Differences were considered statistically significant at *p* < 0.05.

## 3. Results

### 3.1. Comparison of HSPA8 Genotype Characteristics

Patient characteristics are summarized and compared between the groups according to *HSPA8* genotypes (Table 1, Figure 1). There was no difference in sex, age, body weight, or height between NAFLD patients with the major allele or minor allele of *HSPA8* genotype. BMI (*p* = 0.031), prevalence of hypertension (*p* = 0.017), and hyperuricemia (*p* = 0.043) were higher in patients with at least one copy of the minor allele than in those homozygous for the major allele. However, platelet count (*p* = 0.015) and ALT (*p* = 0.015) were higher in patients with the major allele than in those with at least one copy of the minor allele. There were no significant differences in lipid-metabolism-related parameters (total cholesterol, HDL-C, LDL-C, and triglycerides), glucose-metabolism-related parameters (FPG, HbA1c, and HOMA-IR), or renal function (BUN, Cr, and eGFR) between the genotypes. The serum HSC70 concentration was significantly higher in patients with the major allele genotype than in those with at least one copy of the minor allele (9.5 vs. 4.2 mg/L, *p* = 0.013) (Table 1 and Figure 1A). Pathological findings including the prevalence of NASH and severity of steatosis, inflammation, ballooning, and fibrosis did not differ significantly between the genotypes. There was also no significant difference in the Max-IMT between the *HSPA8* genotypes in the patients overall (Table 1 and Figure 1B). 

### 3.2. HSPA8 Genotype Is Associated with Max-IMT in Men

Because there was no relationship between the Max-IMT and *HSPA8* genotypes overall in the patients, we stratified the patients by sex and compared Max-IMT between the genotype groups. In the male patients, the Max-IMT in those with at least one copy of the minor allele was significantly greater than that in patients who were homozygous for the major allele (2.1 vs. 1.2 mm, *p* = 0.005), whereas there was no significant difference among the female patients (1.4 vs. 1.0 mm, *p* = 0.301) (Figure 2A). Max-IMT was divided into three categories (Max-IMT < 1 mm, 1-1.4 mm, and ≥1.5 mm), and patients with at least one copy of the minor allele were compared among the categories. Overall, and in female patients alone, there was no significant difference in the prevalence of the patients with at least one copy of the minor allele compared with that of the patients homozygous for the major allele (*p* = 0.540 and *p* = 0.720) (Figure 2B,C). However, in male patients, the prevalence of patients with at least one copy of the minor allele was greater in those with Max-IMT ≥ 1.5 mm than at the other two Max-IMT levels (68.8% vs. 31.2%, *p* = 0.025) (Figure 2D).

### 3.3. PNPLA3 Genotype Is Not Associated with Max-IMT

To evaluate the association between the *PNPLA3* SNP and atherosclerosis, Max-IMT was compared among the three *PNPLA3* genotypes (Figure 3A). There was no relationship between the Max-IMT and *PNPLA3* genotypes in the patients overall (CC, 1.1 mm; GC, 1.4 mm; and GG, 1.25 mm) (Figure 3A). In the HSPA8 analysis, we next stratified the patients by sex and compared Max-IMT between the genotype groups. However, there was no significant difference in male patients (CC + GC, 1.4 mm; GG, 1.2 mm) or female patients (CC + GC, 1.2 mm; GG, 1.3 mm) (Figure 3B).

### 3.4. Relationship between Hepatic Fibrosis and Arteriosclerosis

The association between arteriosclerosis and age or hepatic fibrosis was investigated. Max-IMT in each hepatic fibrosis stage was evaluated. There was a significant positive correlation between the Max-IMT and age (ρ = 0.4098, *p* < 0.0001) (Figure 4A). The Max-IMT showed a significant difference at different fibrosis stages (*p* = 0.018) (Figure 4B). Moreover, hepatic fibrosis in patients with hypertension was more severe than that in patients without hypertension (*p* = 0.031) (Figure 4C).

### 3.5. Decision-Tree Algorithm for Profiles of Significant Arteriosclerosis

To clarify the profile associated with significant arteriosclerosis (Max-IMT ≥ 1.5 mm) and to evaluate the impact of the *HSPA8* genotype on significant arteriosclerosis, a decision-tree algorithm was generated using four variables (*HSPA8* genotype, age, sex, and hepatic fibrosis) (Figure 5). Age was the first factor that could be associated with significant arteriosclerosis. In patients aged ≥ 65 years, hepatic fibrosis was the second factor to be identified, and 80% of patients with cirrhosis (fibrosis stage = 4) showed significant arteriosclerosis (profile 1). In patients aged ≥ 65 years without liver cirrhosis (fibrosis stages = 0, 1, 2, and 3), sex was the third factor to be identified, and 80% of men and 20% of women showed significant arteriosclerosis. Among them, the *HSPA8* genotype was the fourth factor to be identified in men and women. In men, all patients with at least one copy of the minor allele showed significant arteriosclerosis (profile 2), whereas only 57.1% of the patients homozygous for the major allele showed significant arteriosclerosis (profile 3). In women, 40% of patients with at least one copy of the minor allele (profile 4) and 33.3% of patients with at least one copy of the minor allele (profile 5) showed significant arteriosclerosis. In patients aged < 65 years, sex was the second factor to be identified. In female patients aged < 65 years, *HSPA8* was the third factor, followed by fibrosis stage (profiles 6–9). There were no female patients with significant arteriosclerosis aged < 65 years who were homozygous for the major allele and mild liver fibrosis (stage = 1) (profile 6), whereas 42.9% of these patients showed significant arteriosclerosis if the liver fibrosis stage was 2 or more severe (profile 7). In male patients aged < 65 years, the fibrosis stage was the third factor and *HSPA8* genotype was not a factor (profile 10).

## 4. Discussion

The present study identified the effect of *HSPA8* (rs2236659) SNP on atherosclerosis in NAFLD and clarified sex differences and interactions with confounding factors such as liver fibrosis and age. This work demonstrated that the G allele of the *HSPA8* SNP is associated with a higher prevalence of hypertension, greater Max-IMT in male patients, and significant atherosclerosis of the carotid artery in older male patients. In female patients without advanced liver fibrosis, homozygosity for the major A allele of the *HSPA8* SNP was associated with nonsignificant atherosclerosis. Although NAFLD pathogenesis and its CVD risk are mainly linked to lifestyle-related factors, the current study demonstrates that the genetic background is at least partially related to the ASCVD risk in NAFLD.

Several loci were recently identified as genetic risk factors for NAFLD and CVD in GWAS. For NAFLD, genetic variability in *PNPLA3*, *TM6SF2, MBOAT7, GCKR, HSD17B13*, and other genes was found to be associated with the prevalence of NAFLD and disease progression [32,33,34,35]. *PNPLA3* was also associated with prevalence, pathological severity—including that of liver fibrosis—hepatocarcinogenesis, and mortality in NAFLD patients [21,36,37]. Meanwhile, the associations between these genetic variants including *PNPLA3* and CVD outcome in NAFLD remain unclear. Moreover, in the present study, no association between atherosclerosis and *PNPLA3* was identified (Figure 3). For coronary artery disease, carotid artery atherosclerosis, and peripheral artery disease, numerous loci have been identified as risk factors for disease occurrence and progression [38,39,40,41]. To date, *HSPA8* has not been identified in a GWAS for NAFLD and CVD. Moreover, to the best of our knowledge, no GWAS for CVD in NAFLD has been performed. According to the results of the present study, the *HSPA8* SNP (rs2236659), at least in part, is a specific marker of atherosclerosis in NAFLD. Because CVD is a major event and cause of death in NAFLD patients [4,5,6,7], a GWAS that includes a large cohort should be performed in the near future to elucidate the genetic factors that confer a risk of CVD in NAFLD.

Reactive oxygen species (ROS) and oxidative stress induced by them are key factors involved in the development of CVD and NAFLD. Excessive ROS production reduces the antioxidant capacity and causes further oxidative damage in the liver and vascular wall [42,43]. HSC70 and HSP70 regulate ubiquitin-dependent pexophagy, which selectively removes ROS-stressed peroxisomes, allowing the ubiquitin E3 ligase STIP1 homology and U-box-containing protein 1 to recognize and ubiquitinate ROS-stressed peroxisomes in the cytoplasm, leading to their turnover by autophagy [44]. Therefore, attenuated HSC70 function might cause the development of endothelial cell dysfunction and atherosclerosis. In the present study, at least one copy of *HSPA8* was associated with atherosclerosis in men (Figure 2A,D). Our decision-tree analysis also identified that the minor allele was strongly associated with significant atherosclerosis in noncirrhotic men aged 65 years or older [Figure 5]. Sex is associated with differences in oxidative stress. Under physiological conditions, women are less susceptible than men to oxidative stress due to the antioxidant properties of estrogen and sex differences in NADPH-oxidase activity [45]. Sex differences caused by the *HSPA8* genotype in the present study might be associated with differences in the susceptibility to NAFLD between men and women, and men might be more susceptible than women to developing NAFLD when the minor allele is present.

He et al. reported the association between an *HSPA8* SNP (rs2236659) and coronary heart disease in 2006 Chinese participants, and they tested whether *HSPA8* mRNA expression varied according to mutation status using a luciferase reporter assay with human umbilical vein endothelial cells [24]. In that study, patients who were carriers of the rs2236659 G allele, which is defined as the minor allele in our study, had a decreased risk of coronary heart disease, and the rs2236659 G allele was associated with a 37–40% increase in luciferase expression by the reporter *HSPA8* gene luciferase in endothelial cells compared with that in those with the A allele. In the present study, the serum HSC70 concentration was significantly lower in patients with at least one copy of the rs2236659 G allele than in those without this allele (Figure 1A). Male patients with at least one copy of the rs2236659 G allele showed severe carotid artery atherosclerosis compared with those homozygous for the A allele (Figure 2D). The conclusions of our study and those of the study by He et al. are consistent in showing that *HSPA8*/HSC70 is protective against cardiovascular disease and atherosclerosis. However, the findings on the association between the rs2236659 G allele and *HSPA8*/HSC70 expression did not match, and the clinical phenotypes in patients with the G allele showed conflicting results. A possible explanation for these contradictory results is that the serum HSC70 protein was measured in the present study. The process of *HSPA8* gene transcription into the HSC70 protein occurs in a complex environment, namely, the microenvironment in the body, while also potentially being influenced by the external environment. Therefore, it is important to evaluate the influence of functional protein HSC70 in the real world, not only the gene *HSPA8* as a substance. To the best of our knowledge, tissue- and cell-specific expression of *HSPA8*/HSC70 mRNA and protein has not been well studied. Moreover, all of the patients included in our study also had NAFLD. Compensatory feedback in multiple tissues including the liver might affect the circulating HSC70 protein concentration and might result in a lower serum HSC70 concentration in patients with at least one copy of the G allele than in those with homozygosity of the A allele. Severe atherosclerosis was observed in male patients with at least one copy of the G allele in the present study. Tissue-specific expression of quantitative trait loci and the biological significance of circulating HSC70 should be further investigated to identify the clinical implications of the *HSPA8* SNP.

Chaperone-mediated autophagy (CMA) is a lysosomal-dependent protein degradation pathway. HSC70 specifically binds protein targets and transports them for CMA degradation [46]. Recent studies showed that CMA is critically involved in the development of both NAFLD and CVD [47,48]. Moreover, CMA is associated with the degradation of lipid-droplet-associated proteins and facilitates lipid metabolism in the cell based on the nutrient conditions [49]. Impaired CMA is associated with intracellular lipid accumulation [50] and causes hepatic steatosis [47,49]. Therefore, in the present study, severe hepatic steatosis was expected in the patients with at least with one copy of the minor allele, but there was actually no difference in this regard between those homozygous for the major allele and those with at least one copy of the minor allele (Table 1). Compensatory activation of macrolipophagy, which is independent of HSC70, might cause similar severity of hepatic steatosis between the *HSPA8* genotypes. Our understanding of the effect of the *HSPA8* SNP on NAFLD pathology is currently limited.

In the present study, Max-IMT positively correlated with hepatic fibrosis (Figure 4B), and hepatic fibrosis was an identifiable factor in all of the patients’ profiles (Figure 5). These findings support the results of recent epidemiological studies showing that hepatic fibrosis is a risk factor for CVD events in NAFLD patients [5,7]. However, the *HSPA8* SNP was also an identifiable factor in several patients’ profiles, and there was no association of the *HSPA8* SNP with the hepatic pathological findings, suggesting that it can predict the CVD-specific risk in NAFLD. There are several limitations in the present study. First, the study design was cross-sectional, so the causal relationship between the *HSPA8* SNP and the progression of IMT remains unclear. Moreover, the association between the *HSPA8* SNP and CVD events/mortality also remains unclear. Additionally, this study involved a relatively small sample size recruited at a single center. A multicenter and longitudinal study with a larger sample size is required to confirm the effects of the *HSPA8* SNP on hepatic and CVD pathogenesis in NAFLD patients.

In conclusion, the *HSPA8* SNP (rs2236659) is associated with carotid atherosclerosis in NAFLD in men. To predict the risk of CVD, genotyping the *HSPA8* SNP might be a useful approach, particularly for older male patients.

## Figures and Tables

**Figure 1 genes-13-01265-f001:**
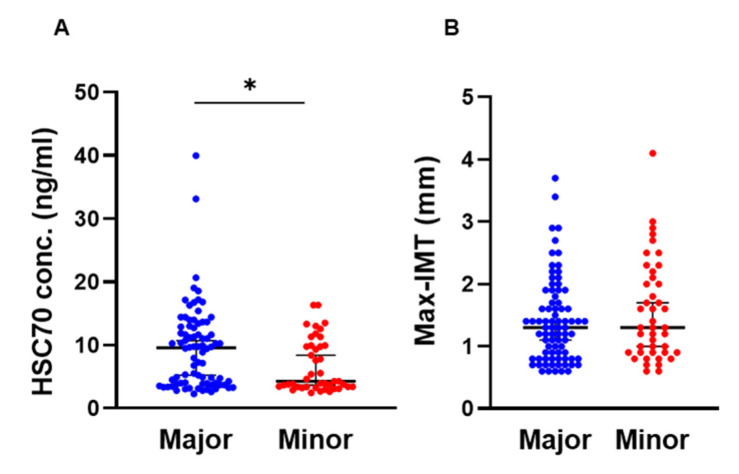
Comparison of serum HSC70 concentration and Max-IMT between *HSPA8* genotypes. (**A**,**B**) HSC70 concentration (**A**) and Max-IMT (**B**) were compared between the patients homozygous for the major allele and those with at least one copy of the minor allele in the patients overall. The middle bar represents the median and the upper/lower bars represent the 95% confidence interval. * *p* < 0.05. HSC70, heat shock cognate 71 kDa protein; Max-IMT, maximum intima–media thickness.

**Figure 2 genes-13-01265-f002:**
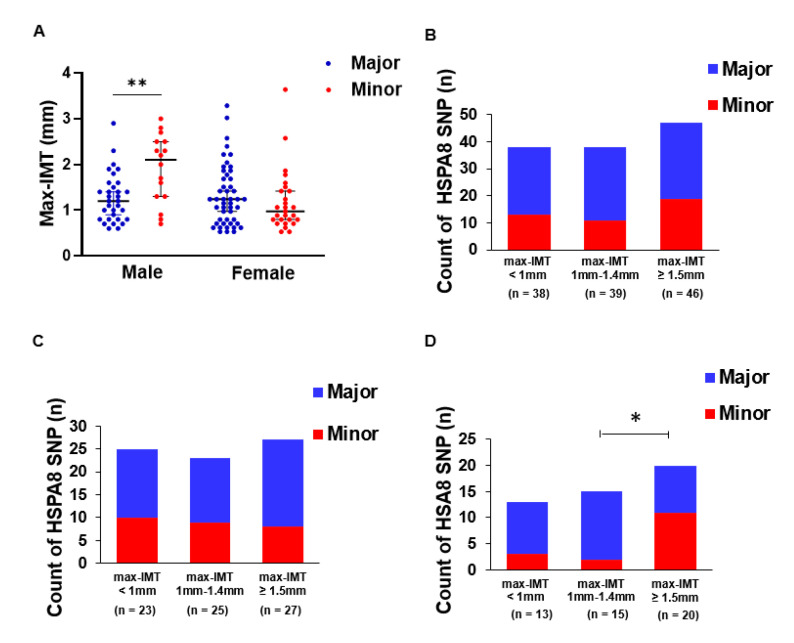
Sex differences in the association of Max-IMT and *HSPA8* genotypes. (**A**) Comparison of Max-IMT between the *HSPA8* genotypes stratified by sex. The middle bar represents the median and the upper/lower bars represent the 95% confidence interval. (**B**–**D**) *HSPA8* genotype prevalence in the individual Max-IMT categories overall (**B**) and in female patients (**C**) and male patients (**D**). * *p* < 0.05 and ** *p* < 0.01. Max-IMT, maximum intima–media thickness; *HSPA8*, heat shock 70 kDa protein 8.

**Figure 3 genes-13-01265-f003:**
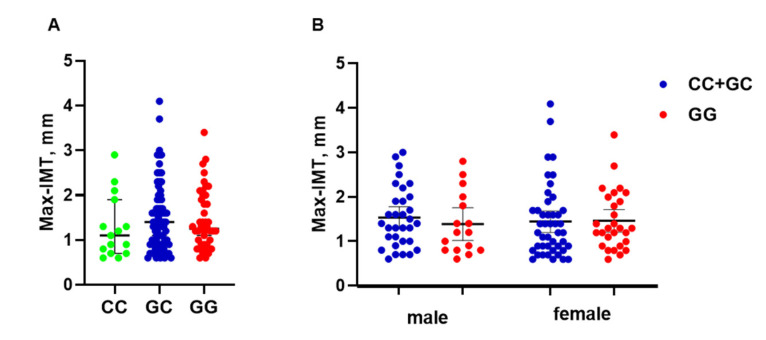
Max-IMT and *PNPLA3* genotypes. (**A**) Max-IMT was compared among three *PNPLA3* genotypes. (**B**) Comparison of Max-IMT between the two *PNPLA3* genotype groups stratified by sex. The middle bar represents the median and the upper/lower bars represent the 95% confidence interval. Max-IMT, maximum intima–media thickness; *PNPLA3*, patatin-like phospholipase domain-containing protein 3.

**Figure 4 genes-13-01265-f004:**
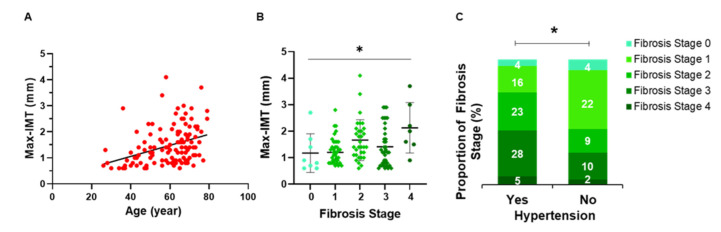
Association among age, liver fibrosis, and Max-IMT. (**A**) Correlation between age and Max-IMT. (**B**) Max-IMT at each hepatic fibrosis stage. The middle bar represents the median and the upper/lower bars represent the 95% confidence interval. (**C**) Association between hypertension and hepatic fibrosis stage. * *p* < 0.05. Max-IMT, maximum intima–media thickness.

**Figure 5 genes-13-01265-f005:**
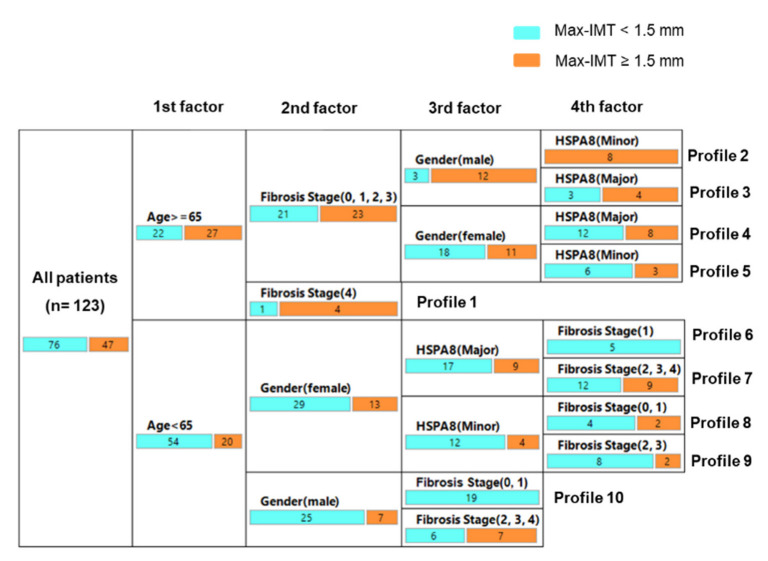
Decision-tree analysis to identify the factors associated with significant Max-IMT. Max-IMT, maximum intima–media thickness.

**Table 1 genes-13-01265-t001:** Characteristics of patients according to presence of *HSPA8* SNP.

	Major Allele	Minor Allele	*p* Value
Characteristics	n = 80	n = 43	
Gender, Male/Female, n (%)	32 (40)/48 (60)	16 (37.2)/27 (62.8)	0.762
Age, Years	60 (47–69)	64 (56–68)	0.173
Height, cm	157.5 (151.2–165.3)	156.8 (150.9–163.5)	0.392
Weight, kg	72.2 (60.8–78.5)	73.5 (63.2–83.7)	0.282
BMI, kg/m^2^ *	27.5 (25.6–32)	30 (28–34.1)	0.031
Hypertension, n (%) *	43 (53.8)	33(76.7)	0.012
Dyslipidemia, n (%)	63 (78.8)	36 (83.7)	0.507
Diabetes, n (%)	49 (61.3)	31 (72.1)	0.229
Hyperuricemia, n (%) *	26 (32.5)	22 (51.2)	0.043
Platelet count, ×103/µL	19.9 (16.2–23.5)	17.2 (14.9–21.3)	0.015
Albumin, g/dL	4.1 (3.8–4.3)	4 (3.8–4.2)	0.512
AST, U/L	56.5 (38–77.8)	50 (29–78)	0.442
ALT, U/L *	63 (40.8–103.5)	47 (31–72)	0.015
GGT, U/L	63.5 (43.3–111.5)	65 (39–93)	0.384
ALP, U/L	242 (191.8–301)	224 (190–263)	0.176
Total cholesterol, mg/dL	187 (162–212.8)	175 (160–198)	0.231
HDL-C, mg/dL	47 (39.3–60.8)	47 (40–57)	0.554
LDL-C, mg/dL	116.5 (94–135.3)	104 (93–127)	0.283
Triglyceride, mg/dL	145.5 (106.5–178)	151 (113–187)	0.494
BUN, mg/dL	13.3 (11.2–16.4)	14.5 (11.7–16.9)	0.323
Cr, mg/dL	0.69 (0.6–0.9)	0.69 (0.6–0.9)	0.775
eGFR, ml/min	75.1 (63.2–89.7)	67.3 (57.2–91.1)	0.259
FPG, mg/dL	109 (97–134)	112 (95–130)	0.920
HbA1c, %	6.2 (5.6–7.3)	6.1 (5.7–6.6)	0.696
HOMA-IR	4.3 (2.8–7.1)	5 (3.2–6.9)	0.392
HSC70 (ng/mL) *	9.53 (3.9–13)	4.2 (3.3–9.9)	0.013
Max-IMT, mm	1.3 (0.8–1.8)	1.3 (0.9–2.1)	0.392
NAFL/NASH, n	18/62	13/30	0.346
Steatosis Score, n (0/1/2/3)	0/52/18/10	0/30/9/4	0.828
Inflammation Score, n (0/1/2/3)	2/42/30/6	2/25/13/3	0.803
Ballooning Score, n (0/1/2)	17/35/28	14/16/13	0.387
Fibrosis Stage, n (0/1/2/3/4)	3/29/20/24/4	5/9/12/14/3	0.274

* Indicates that the data are statistically significant difference (*p* < 0.05). Data are expressed as median (range) or number. Abbreviations: M/F, male/female; BMI, body mass index; AST, aspartate aminotransferase; ALT, alanine aminotransferase; GGT, γ-glutamyl transpeptidase; ALP, alkaline phosphatase; HDL-C, high-density-lipoprotein cholesterol; LDL-C, low-density-lipoprotein cholesterol; BUN, blood urea nitrogen; Cr, creatinine; HOMA-IR, Homeostatic Model Assessment for Insulin Resistance; HSC70, heat shock cognate protein 70; Max-IMT, the maximum of all local maximal measures of intima–media thickness; eGFR, estimated glomerular filtration rate; FPG, fasting plasma glucose; HbA1c, hemoglobin A1c; NAFL/NASH, nonalcoholic fatty liver/nonalcoholic steatohepatitis.

## Data Availability

The data presented in this study are available on request from the corresponding author. The data are not publicly available due to privacy.

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
