# Peer review of "HSPA8* Single-Nucleotide Polymorphism Is Associated with Serum HSC70 Concentration and Carotid Artery Atherosclerosis in Nonalcoholic Fatty Liver Disease"

_genes, 2022, doi:10.3390/genes13071265_

Round 1
Reviewer 1 Report
Overall, the idea of the work is interesting. The findings of the work could have merit in the related field. Just, some concerns should be addressed.
General comment
The authors should ensure that all gene names throughout the manuscript are italicized to match the standards of HUGO for gene nomenclatures.
Introduction
Line 64: please spell out the “PNPLA3”.
- Line 71: “p.I148M, and they identified that M variant heterozygosity….” the authors should keep consistency in their elaboration regards the gene variants either at the level of the DNA by mentioning the reference sequence of the variant or at the protein level. Please specify the type of this variant at the DNA level to facilitate the readers follow the authors in their elaboration.
- Lines 72 and 73:” Thus, the association between the PNPLA3 genotype and CVD in NAFLD has not been confirmed” this sentence supports that the authors should investigate the relation of this gene variants with the NAFLD as long as there is controversy in the previous publications. Why did the authors elaborate too much in this issue as long as they did not include this gene in their study? This section of the introduction needs reconstruction to give more examples for other genetic variants could be related to NAFLD and to generalize the findings of the previous studies then concentrate on the specified gene in this work.
Methods
- Lines 116 and 118 please revise and delete the data duplication.
- The authors should provide the minimum detection level, the inter- and the intra-assay coefficient of variations for the assessed HSC70 using the enzyme linked immuno-sorbent assay as part of the applied quality control measures.
- Did the authors measure the concentration and quality of the extracted DNA?
- Thanks to the authors for providing the primer sequences applied in their PCR. Did they design or extract these sequences from previous publications?
- What were the quality control measures the authors apply in their genotyping technique?
Results
They are representative for the findings just the authors should improve the resolution of the provided figures in particular Figure 2 as It was very hard to identify the x-axis labels.
Author Response
Thank you for the positive comments and suggestions. We revised the manuscript accordingly, which we believe markedly improved our manuscript. Response to the comment was attached.

Reviewer 2 Report
The manuscript” HSPA8 single nucleotide polymorphism associated with serum HSC70 concentration and carotid artery atherosclerosis in non-alcoholic fatty liver disease” by Zhao et al. elucidates the linkages among rs2236659 of HSPA8, atherosclerosis, and hepatic pathogenesis in NAFLD. The observation of this type would be highly significant even though the present study is based on a small population of patients. Overall, the manuscript is well-written and organized and it will be better to present the figures with higher resolution and larger font size to read. Also, it will make this a more exciting study if the author can analyze the SNP of other genes such as PNPLA3 (rs738409) to tell out if rs2236659 of HSPA8 is a specific marker or master disease-causing gene. I am also interested in other SNPs of HSPA8 with confusion about why the author only focuses on rs2236659.
Author Response
Thank you for the positive comments and valuable suggestions. We revised the manuscript accordingly. We believe that this markedly improved the manuscript. Response to the comment was attached.

Round 2
Reviewer 1 Report
The authors have adequately addressed the concerns raised by the reviewer. Thank you